# Panoptic Image Segmentation Method Based on Dynamic Instance Query

**DOI:** 10.3390/s25092919

**Published:** 2025-05-05

**Authors:** Lanshi Yang, Shiguo Wang, Shuhua Teng

**Affiliations:** 1School of Computer Science and Technology, Changsha University of Science and Technology, Changsha 410076, China; 22408050281@stu.csust.edu.cn (L.Y.); sgking@csust.edu.cn (S.W.); 2School of Electronic Information, Hunan First Normal University, Changsha 410205, China

**Keywords:** deep learning, image segmentation, instance segmentation, panoptic segmentation, dynamic instance queries

## Abstract

Panoptic segmentation, as a key task in the field of computer vision, holds significant importance in practical applications such as autonomous driving and robot vision. Currently, among deep-learning-based panoptic segmentation methods, query-based methods have received widespread attention. However, existing methods, such as Mask2Former, typically rely on a static query mechanism. This makes it difficult for the model to adapt to changes in the number of instances in different scenes and can lead to instance loss or confusion, thus limiting performance in complex scenes. Furthermore, it is prone to insufficient feature extraction and a loss of global information. To address these problems, this paper proposes a panoptic segmentation method based on dynamic instance queries (PSM-DIQ). PSM-DIQ uses a multi-dimensional attention mechanism to enhance feature extraction, utilizes instance-activation-guided dynamic query generation to improve the ability to distinguish between different instances, and optimizes pixel–query interactions through a dual-path Transformer decoder. Experiments on the Cityscapes and MS COCO datasets show that, based on the ResNet-50 backbone, PSM-DIQ significantly outperforms the Mask2Former baseline, with PQ values improving by 1.8 and 1.7 percentage points, respectively. The experimental results verify the effectiveness of PSM-DIQ in complex scene panoptic segmentation. Finally, this work will be released as an open-source software package on GitHub (v1.0).

## 1. Introduction

Visual sensors, as crucial components in modern intelligent systems, play a core role in fields such as autonomous driving, robot vision, and intelligent surveillance. With the development of new visual sensors like panoptic cameras, efficiently processing and understanding large-scale visual sensor data has become an important challenge. The goal of computer vision is to enable machines to possess human-like visual capabilities, thus achieving tasks such as image processing, scene parsing, and the 3D reconstruction of sensor-acquired images. As a key technology for visual sensor data processing, image segmentation [1] has long been a research hotspot, mainly covering directions such as semantic segmentation [2], instance segmentation [3], and panoptic segmentation [4]. Traditional semantic segmentation can classify each pixel acquired by the sensor, but it cannot distinguish different instances within the same category. Instance segmentation, while able to identify and segment independent objects in the foreground, usually ignores background information. These limitations make it difficult for traditional methods to comprehensively describe the scene content captured by visual sensors. In contrast, panoptic segmentation, by combining the advantages of semantic segmentation and instance segmentation, analyzes visual sensor data from a global perspective, ensuring that each pixel not only possesses a semantic label but is also distinguishable from other pixels of the same category, thereby providing a more comprehensive scene understanding capability.

Panoptic segmentation, as a crucial task in computer vision, has undergone rapid development since its proposal by Kirillov et al. [4]. Based on technical approaches, existing panoptic segmentation methods can be primarily categorized into three classes: top-down, bottom-up, and query-based methods. Top-down approaches, such as Panoptic FPN [5] and UPSNet [6], typically perform instance segmentation and semantic segmentation separately before merging results through post-processing strategies. However, these methods often exhibit inconsistencies at boundaries and lack true end-to-end training. Bottom-up methods like Panoptic-DeepLab [7] and Deeperlab [8] first predict per-pixel categories and center offsets and then generate instances through clustering. While avoiding post-processing fusion issues, such approaches struggle with occlusion and complex scenes. With the application of Transformers in computer vision, DETR [9] pioneered the integration of attention mechanisms with queries for object detection. Subsequent query-based methods like MaskFormer [10] and Mask2Former [11] extended this to segmentation tasks, achieving genuine end-to-end panoptic segmentation by learning query-feature interactions for instance prediction. These methods have demonstrated outstanding performance on benchmarks like MS COCO and Cityscapes, currently dominating the field. Nevertheless, most existing query-based approaches employ static query mechanisms, and their performance remains constrained by inherent limitations, particularly when handling complex scenes and dynamically changing sensor data.

Query-based methods, with their end-to-end training approach and excellent sensor data processing capabilities, have emerged as a prominent area in panoptic segmentation research. These methods transform the panoptic segmentation problem into a query matching and decoding process, capturing global and local context information through interactions between queries and sensor feature maps. Query-based panoptic segmentation approaches typically require the manual construction of initial query variables as input, where each “instance mask” and “semantic mask” in the final segmentation results must correspond to a separate query vector. This makes the determination of the initial query quantity a critical challenge. However, static query methods represented by Mask2Former [11] exhibit significant limitations when processing dynamic visual sensor data: firstly, static queries employ preset fixed quantities (typically 100 or 200), which perform poorly when handling varying numbers of targets across different scenarios. The actual number of instances in input images is typically unknown in advance-insufficient query quantities, leading to instance loss, while excessive quantities create unnecessary overhead. For instance, in complex urban scenes from the Cityscapes dataset (averaging 25 instances/image), 75% of queries are wasted when actual instances are far fewer than preset queries; in complex scenarios from the MS COCO dataset, where instance counts approach or exceed preset values (over 100 instances in some images), this leads to sensor-captured instance loss/confusion (as shown in Figure 1), even causing single queries to correspond to multiple instance masks. Secondly, static queries remain completely independent of sensor input, relying solely on learned fixed-parameter representations, severely limiting model adaptability to different scene characteristics. Finally, static queries require iterative optimization through multiple decoder layers to achieve good representations. This not only increases computational complexity but also extends inference time, adversely affecting the real-time performance of visual sensor systems—particularly detrimental for time-sensitive applications like autonomous driving.

Comparative analysis of existing panoptic segmentation methods reveals distinct advantages and limitations across different technical approaches: top-down approaches benefit from a modular design, enabling the flexible integration of advanced semantic and instance segmentation networks. However, their independent fusion processes hinder holistic performance optimization, often causing inconsistencies at category boundaries. Bottom-up methods demonstrate high computational efficiency and better sensitivity to small targets, yet struggle with occlusion handling and separating adjacent instances. Static-query-based methods achieve superior global understanding through end-to-end training and attention mechanisms. Nevertheless, as previously discussed, fixed query quantities, query-input independence, and high computational costs remain their primary bottlenecks. Particularly in visual sensor applications, the diversity of scene complexity and target types demands segmentation algorithms with enhanced adaptability and efficiency. For example, autonomous driving systems must simultaneously process targets of varying scales and quantities (roads, pedestrians, vehicles), and surveillance systems require the real-time processing of massive data under resource-constrained conditions. These application-specific characteristics further expose the limitations of existing methods while providing clear directions for improvement in this study: developing a panoptic segmentation approach capable of dynamically adapting to scene complexity, maintaining computational efficiency, and ensuring accuracy.

To address the shortcomings of static query methods in visual sensor data processing, this paper proposes a panoptic segmentation network with dynamic instance queries (PSM-DIQ). Our core concept involves “dynamically” generating required query vectors by analyzing the input image itself: first obtaining basic information (instance count, approximate locations) through simple methods, then constructing query vectors based on this information, and finally refining the panoptic segmentation results through Transformer architecture. Unlike previous works, our method eliminates dependence on predefined query quantities by dynamically generating scene-complexity-adaptive queries through image content analysis. Moreover, since attention mechanisms fundamentally generate corresponding masks by computing the similarity between query vectors and instance high-dimensional embeddings, we specifically design mechanisms that ensure that dynamic queries inherently possess similarity with instance embeddings, as well as sufficient discriminability between different query vectors. The network enhances sensor data processing through three innovative designs: (1) the introduction of a multi-dimensional attention mechanism that fuses spatial and channel information to enhance the feature extraction capability of visual sensors; (2) a proposed dynamic query method guided by instance activation that selects semantically rich feature embeddings from the sensor’s feature map, alleviating the burden on the Transformer decoder and improving both the efficiency of the query and its relevance to the sensor data; (3) the design of a dual-path Transformer decoder architecture that accurately captures image details and semantic information collected by the sensor by alternately updating query features and pixel features. The experiments show that our method achieves excellent performance on multiple public datasets, significantly improving the processing efficiency and accuracy of visual sensor data. These innovations are of great significance to promoting the development of intelligent sensor systems.

## 2. Method

### 2.1. Network Infrastructure

To address the inherent limitations of static query methods in visual sensor data processing, we propose a panoptic segmentation network based on dynamic instance queries (PSM-DIQ). This section elaborates on the design concept of the network architecture and the intrinsic connections between its components.

#### 2.1.1. Overall Architecture Design

The PSM-DIQ network adopts an end-to-end design concept and consists of four closely related key modules: the backbone network, the pixel decoder, the query generation module based on instance activation, and the dual-path Transformer decoder, as shown in Figure 2. These modules form a complete processing pipeline that realizes the transformation from raw sensor data to accurate segmentation results in a layer-by-layer progressive manner. Different from traditional static methods [12], our method introduces three innovative components:

(1) **CBAM attention mechanism:** Traditional backbone networks often fail to effectively capture key features when processing complex scenes. To address this problem, we introduce the CBAM to enhance the feature extraction capability of visual sensors. This module enables the network to adaptively focus on important areas and channels by simultaneously considering the attention distribution in spatial and channel dimensions. This multi-dimensional fusion mechanism is particularly suitable for processing high-dimensional data streams generated by panoptic vision sensors, laying a solid feature foundation for subsequent dynamic query generation;

(2) **Instance Activation Guidance Module:** The number of queries preset by static query methods cannot adapt to changes in scene complexity, resulting in a waste of computing resources or insufficient expression capabilities. The instance activation guidance module that we proposed dynamically generates queries from the low-level feature maps of the sensor and adaptively determines the required number of queries and initial features by analyzing the image content. This data-driven approach significantly improves the relevance of queries to sensor data and provides a flexible adaptation mechanism for the ever-changing scene complexity in real-time sensor applications;

(3) **Dual-path decoder module:** Traditional decoders often have difficulty balancing global semantic understanding and local detail preservation. Our dual-path Transformer decoder effectively resolves this contradiction by alternately updating and optimizing sensor pixel features and query features. This design is specifically designed for complex scene information captured by visual sensors and can simultaneously maintain high-level semantic consistency and fine-grained boundary accuracy.

#### 2.1.2. Information Flow Processing

The processing flow of PSM-DIQ embodies the progressive extraction and integration from low-level features to high-level semantics. As shown in Figure 2, the image captured by the sensor is first input into the backbone network, and multi-scale feature extraction is performed through the CBAM attention mechanism. This mechanism is specially designed to handle various sensor input conditions (such as illumination changes and scene complexity) and generate three feature maps, E3, E4, and E5, with a resolution of 1/8, which effectively captures sensor information of different scales.

These feature maps are then mapped to 256-channel feature maps through a 1 × 1 convolutional layer and input into the pixel decoder. As a feature optimization step, the pixel decoder is responsible for aggregating contextual information and outputting enhanced multi-scale feature maps $E3^{\prime }$, $E4^{\prime }$, and $E5^{\prime }$. This design ensures the efficient processing of high-resolution sensor data while maintaining computational efficiency. This step provides informative feature representations for subsequent dynamic query generation.

Based on the obtained feature representation, the network performs an instance-activation-guided query strategy from feature map E4 to dynamically generate $N_{a}$ object queries. These queries are then concatenated with $N_{b}$ context-assisted queries (consisting of context- and image-independent information) to form the total query Q. This dynamic query generation method is particularly suitable for real-time sensor applications where the scene content changes rapidly, solving the limitation of a fixed number of queries in static query methods.

Finally, the dual-path Transformer decoder takes the total query Q and the flattened high-resolution pixel features $E3^{\prime }$ as input. As shown in Figure 2, the decoder uses a dual-path approach to update the pixel features $E3^{\prime }$ and the query Q, which is optimized for efficient sensor data processing. The object category and segmentation mask are predicted in each decoder layer, realizing the transformation from features to the final segmentation result.

The following subsections describe the backbone network, pixel decoder, instance-guided query module, dual-path decoder, and loss function of PSM-DIQ in more detail, with special emphasis on their roles and their intrinsic connections in processing and analyzing visual sensor data.

### 2.2. Attention-Mechanism-Based Backbone Network

The attention mechanism enhances the model’s perceptual capabilities by focusing on specific, relevant information in the image and suppressing irrelevant information. SENet [13], ECANet [14], and VoVNet [15], which are based on the attention mechanism, all show better performance compared to their original counterparts. In recent years, the attention mechanism has been widely used in the field of image segmentation, improving segmentation results by focusing on relevant feature information in the image. To address the problem of poor instance and semantic segmentation results caused by inadequate feature extraction, this paper introduces channel attention and spatial attention in the downsampling stage, learning the correlation between channels and spatial locations and highlighting the edge details of lower-resolution features, thereby achieving the precise segmentation of semantic and instance classes and improving segmentation accuracy.

In the field of image processing, channel attention and spatial attention mechanisms are widely used in neural network structures. Each focuses on the channel information and spatial information of the features, respectively, but, when used alone, they may lead to the loss of spatial or channel information. To obtain more complete feature information, a convolutional block attention module (CBAM), which combines channel and spatial attention, is introduced to better capture the channel and spatial information of the features, thereby improving the prediction of semantic and instance classes and increasing the accuracy of panoptic segmentation. CBAM, as shown in Figure 3, includes a channel attention module and a spatial attention module, which generate attention feature maps sequentially in the channel and spatial dimensions. These attention maps are then multiplied with the input feature map to adaptively refine the features and generate the final feature map. The mathematical expression of the CBAM can be formalized as follows:(1)$$F^{\prime }=M_{c}\left(F\right)⊗F$$(2)$$F^{\prime \prime }=M_{s}\left(F^{\prime }\right)⊗F^{\prime }$$

Among them, *F* represents the input feature map, $M_{c}$ and $M_{s}$ represent the channel attention module and the spatial attention module, respectively, ⊗ represents the element-wise multiplication operation, and $F^{\prime \prime }$ is the final enhanced feature map.

The channel attention module focuses on extracting the channel information of the feature map, and enhances the effective feature channels and suppresses the invalid channels by learning the importance weights of different channels. The specific implementation process is as follows:(3)$$M_{c}\left(F\right)=\sigma \left(MLP\left(A\nu gPool\left(F\right)\right)+MLP\left(MaxPool\left(F\right)\right)\right)$$

Among them, AνgPool and MaxPool represent global average pooling and global maximum pooling operations, respectively, MLP represents a multi-layer perceptron consisting of two fully connected layers, and σ represents a sigmoid activation function. By combining the information of average pooling and maximum pooling, the channel attention module can more comprehensively capture the importance of channel-level features. The calculation process of channel attention can be further expanded as follows:(4)$$\begin{array}{cc} \; & F_{a\nu g}^{c}=\frac{1}{H\times W}\sum_{i=1}^{H}\sum_{j=1}^{W}F\left(i,\;j\right)\\ \; & F_{max}^{c}=\max_{i,j}F\left(i,\;j\right)\\ \; & M_{c}\left(F\right)=\sigma \left(W_{1}\left(W_{0}\left(F_{avg}^{c}\right)\right)+W_{1}\left(W_{0}\left(F_{max}^{c}\right)\right)\right) \end{array}$$
where *H* and *W* represent the height and width of the feature map, respectively, and $W_{0}$ and $W_{1}$ are the weight parameters in the MLP. The spatial attention module focuses on the spatial position information of the feature map and focuses on the key areas by learning the importance of different spatial positions. The implementation process is as follows:(5)$$M_{s}\left(F\right)=\sigma \left(f^{7\times 7}\left(\left[AvgPool\left(F\right);\;MaxPool\left(F\right)\right]\right)\right)$$
where $f^{7\times 7}$ represents a convolution layer with a kernel size of 7×7, and [;] represents a concatenation operation in the channel dimension. The spatial attention module generates spatial attention weights by concatenating the average pooling and maximum pooling results in the channel dimension and then applying a convolution operation.

### 2.3. Pixel Decoder

The pixel decoder is a neural network component used in image processing and computer vision tasks. Its main function is to downsample and combine high-resolution feature maps, thereby generating lower-resolution features with more semantic information, providing input for the subsequent multi-task decoder. Pixel decoders are usually based on the feature pyramid network (FPN) architecture. The FPN is able to integrate features from different levels of the backbone (such as ResNet, Swin Transformer), which typically contain different spatial resolutions and semantic information. The role of the FPN is to fuse high-level semantic features with low-level detail features to generate multi-scale features. Multi-scale contextual features are crucial for image segmentation. However, using complex multi-scale feature pyramid networks increases the computational burden. Unlike previous methods [11] that directly use feature maps from the pixel decoder, we use the refined pixel features from the Transformer decoder (extracted through the attention mechanism described in Section 3.2) to generate segmentation masks. This setting reduces the pixel decoder’s need for extensive context aggregation. Therefore, we can use a lightweight pixel decoder module. To better balance accuracy and speed, this section utilizes a PPM-FPN [16] variant to leverage the refined pixel features produced by the Transformer decoder rather than relying on the basic feature maps from the traditional pixel decoder (the rationale for selecting the PPM-FPN is demonstrated in the extended experiments in Section 3.5).

### 2.4. Instance-Activation-Guided Queries

Inspired by DETR, this section proposes a novel instance-activation-guided query (IA-guided) method. Unlike the learnable queries used in DETR, IA-guided queries are not randomly initialized or globally learned but are instead derived by directly selecting pixel embeddings with high semantic activation from the underlying multi-scale feature map to serve as queries. This design overcomes the limitations of traditional learnable queries, which are independent of the input image content and may lead to query redundancy or information loss. By utilizing the semantic information in the underlying feature map, IA-guided queries can more efficiently encode the target information in the image, reduce the burden on the Transformer decoder, and improve the queries’ correlation with the image content.

From the perspective of information theory, the prior information of static queries (including learnable queries) is almost zero because they are independent of the input image. This means that the model must rely entirely on the multi-layer self-attention mechanism of the Transformer decoder to gradually associate the queries with the target instances. This process can lead to two problems:**Information redundancy:** Most static queries may focus on background areas or unimportant targets, resulting in wasted computational resources.**Information loss:** A few static queries may not be able to capture all important target instances, especially small or occluded targets.

In contrast, the prior information of IA-guided queries comes from the semantic information of the underlying feature map. Using an auxiliary classification head, we can predict the probability that each pixel belongs to the foreground. This means that IA-guided queries already “know” which areas in the image are more likely to contain targets in the initialization stage. Therefore, IA-guided queries have higher initial information entropy and can more effectively encode the target information in the image. This enables the model to more effectively utilize limited query resources, reduce redundant calculations, and improve sensitivity to targets.

#### 2.4.1. Auxiliary Classification Head

This method designs an auxiliary classification head structure to optimize the object query generation process (as shown in Figure 4). Specifically, an auxiliary classification head is added on feature map $E_{4}$ to generate a per-pixel category probability prediction. The structure of the auxiliary classification head consists of two convolutional layers:(6)$$\begin{array}{cc} F_{1} & ={\mathrm{Conv}}_{3\times 3}\left(E_{4}\right)\\ P & ={\mathrm{Conv}}_{1\times 1}\left(F_{1}\right) \end{array}$$

These two convolutional layers together form a lightweight semantic segmentation network for extracting semantic information from the underlying feature map. The first convolutional layer uses a 3×3 kernel, primarily for capturing local features and enhancing the model’s spatial awareness. The second convolutional layer uses a 1×1 kernel, which integrates local features and generates the final category probability prediction.

Through this structural design, the auxiliary classification head can effectively extract information that is helpful for classification and generate more accurate category probability predictions. During the category probability prediction process, the category probability prediction $p_{i}\in \Delta^{K+1}$ of each pixel is calculated using the maximum activation value. This probability distribution reflects the likelihood of each pixel belonging to different categories (including the background), where $\Delta^{K+1}$ is the (K+1)-dimensional probability simplex, *K* is the number of target categories, and the additional 1 dimension represents the “no object” category. This design ensures that the model not only identifies the target category but also distinguishes the background area, thereby improving prediction accuracy.

#### 2.4.2. Activation Query Selection

Furthermore, we determine the foreground probability $p_{i,k_{i}}$ of each pixel, where $k_{i}$ is the category index corresponding to the maximum activation value:(7)$$k_{i}=\underset{k\in \{1,2,\cdots ,K\}}{\arg\max}\;p_{i,k}$$

To reduce computational complexity, we simplify the problem to a binary classification task (foreground or background) by selecting the category index corresponding to the maximum activation value. Based on these foreground probabilities, pixel embeddings with higher foreground probabilities are selected from feature map $E_{4}$ as the foundation for object queries and, finally, $N_{a}$ pixel embeddings are selected for object queries.

In this method, we select pixels with local maximum foreground probability in their respective category plane. The selection method for local maximum foreground probability involves examining the 8-neighborhood of a pixel to find the pixel with the highest foreground probability. We define the spatial 8-neighborhood index set of pixel *i* as δ(i). If the foreground probability $p_{i,k_{i}}$ of pixel *i* is greater than or equal to the foreground probability of all pixels in its neighborhood, the pixel is considered as a local maximum:(8)$$\mathrm{LocalMax}\left(i\right)=\left\{\begin{array}{cc} 1, & \mathrm{if}\;p_{i,k_{i}}\geq p_{j,k_{j}}\;\mathrm{for}\;\mathrm{all}\;j\in \delta \left(i\right)\\ 0, & \mathrm{otherwise} \end{array}\right.$$

After selecting pixels with local maximum foreground probability, we further select from them the pixels with the highest global foreground probability to serve as the final object queries. This layer-by-layer screening method ensures that the selected pixels are not only locally representative but also better reflect the target’s feature information globally:(9)$$Q=\mathrm{TopK}(\left\{E_{4}\left[i\right]|\mathrm{LocalMax}\left(i\right)=1\right\},N_{a})$$
where the TopK function selects the $N_{a}$ local maximum pixels with the highest foreground probability $p_{i,k_{i}}$.

Since non-local-maximum probability pixels have pixels with higher scores already selected within their neighborhood, these pixels are prioritized, which further improves the accuracy of object queries. This dual screening mechanism ensures that the selected pixels are both locally representative (with the highest foreground probability in the neighborhood) and globally significant (with the highest foreground probability among all local maximum pixels).

#### 2.4.3. Matching Mechanism Innovation

Traditional object detection and segmentation matching typically rely on predefined anchor boxes or region proposals and use binary classification scores to evaluate matching quality. This approach requires extensive post-processing operations (such as non-maximum suppression) when handling complex scenes and struggles to adapt to targets with varying shapes.

Our method innovatively changes the matching strategy by not relying on prior anchor boxes or binary classification scores but instead using only the category prediction and a simple location cost function $L_{loc}$ to calculate the assignment cost. The location cost $L_{loc}$ is defined as follows:(10)$$L_{loc}\left(i,g\right)=\left\{\begin{array}{cc} 0, & \mathrm{if}\;\mathrm{pixel}\;i\;\mathrm{is}\;\mathrm{located}\;\mathrm{within}\;\mathrm{target}\;\mathrm{object}\;g\\ 1, & \mathrm{otherwise} \end{array}\right.$$
where *i* represents the pixel position on the feature map and *g* represents the ground truth target object.

This location cost design ensures that only pixels falling inside a target object participate in the matching process, thereby improving the accuracy of category and mask predictions. The final matching cost can be represented as follows:(11)$$L\left(i,\;g\right)=\lambda_{cls}\cdot L_{cls}\left(i,\;g\right)+\lambda_{loc}\cdot L_{loc}\left(i,\;g\right)$$
where L(i, g) represents the total matching cost between pixel *i* and target *g*, $L_{cls}\left(i,\;g\right)$ represents the category prediction loss, $L_{loc}\left(i,\;g\right)$ is the location cost, and $\lambda_{cls}$ and $\lambda_{loc}$ are the balancing weights for the category loss and location loss, respectively.

### 2.5. Dual-Path Transformer Decoder

In traditional Transformer decoders, a single-path update strategy is typically employed, which involves updating either pixel embeddings or query embeddings. This unidirectional information flow may lead to the insufficient utilization of information, limiting the model’s expressive power. To address this issue, this section proposes a dual-path update strategy as shown in the dotted box on the right side of Figure 2, inspired by the expectation–maximization (EM) algorithm. This strategy alternately updates pixel features and queries, enabling them to exchange information more comprehensively, thereby achieving a more refined feature representation and improving the model’s learning efficiency and prediction accuracy.

In the dual-path update strategy, the target instance (or background) to which each pixel belongs can be viewed as a latent variable. Pixel feature updates and query updates correspond to the E-step and M-step of the EM algorithm, respectively:Pixel Feature Update (E-step): Given the current queries (which can be considered as representations of target instances), the similarity between each pixel feature and each query is calculated using a cross-attention mechanism. This similarity can be interpreted as the posterior probability of a pixel belonging to each target instance. Subsequently, based on these posterior probabilities, the pixel features are updated as a weighted average of the queries. The mathematical expression is as follows:(12)$$A_{ij}=\frac{\exp(Q_{i}\cdot K_{j}/\sqrt{d})}{\sum_{k}\exp\left(Q_{i}\cdot K_{k}/\sqrt{d}\right)}$$(13)$$X^{\prime }=X+\mathrm{CrossAttention}\left(X,Q,V\right)=X+\mathrm{softmax}\left(\frac{QK^{T}}{\sqrt{d}}\right)V$$
where *X* is the pixel feature, *Q*, *K*, and *V* are the query, key, and value matrices generated from the queries, and *d* is the feature dimension.Query Update (M-step): Given the current pixel features and their posterior probabilities, each query is updated using masked attention and self-attention mechanisms. Masked attention restricts the query’s attention range to the foreground region of the mask predicted by the previous layer; this is equivalent to weighting the query based on the pixel’s posterior probability. Self-attention allows queries to interact with each other, further optimizing the query representation. The mathematical expression is as follows:(14)$$\begin{array}{cc} Q^{\prime } & =Q+\mathrm{MaskedAttention}(Q,X,M)+\mathrm{SelfAttention}(Q)\\ \; & =Q+\mathrm{softmax}\left(\frac{Q{\left(XM\right)}^{T}}{\sqrt{d}}\right)X+\mathrm{softmax}\left(\frac{QQ^{T}}{\sqrt{d}}\right)Q \end{array}$$
where *M* is the segmentation mask predicted by the previous layer, restricting the scope of attention.

Through this alternating update scheme, the dual-path update strategy gradually optimizes pixel features and queries, allowing them to mutually adapt and ultimately reach a local optimum. This parallels the EM algorithm’s iterative optimization between latent variables and model parameters, ultimately converging to the maximum likelihood estimate.

#### 2.5.1. Advantages of Dual-Path Updates

The dual-path update strategy significantly enhances the capture of image details and semantic information through the synergistic effect of pixel feature updates and query updates:

**(1) Pixel Feature Updates Enhance Local Detail Capture:** In the E-step, each pixel feature is updated based on its similarity with various queries. This process can be represented as follows:(15)$$X_{l}^{\left(i\right)}=\sum_{j=1}^{N}\alpha_{ij}Q_{l-1}^{\left(j\right)}$$
where $\alpha_{ij}$ represents the attention weight between pixel *i* and query *j*, calculated as follows:(16)$$\alpha_{ij}=\frac{\exp({\left(X_{l-1}^{\left(i\right)}+P_{x}^{\left(i\right)}\right)}^{T}\left(Q_{l-1}^{\left(j\right)}+P_{q}^{\left(j\right)}\right)/\sqrt{d_{k}})}{\sum_{k=1}^{N}\exp\left({\left(X_{l-1}^{\left(i\right)}+P_{x}^{\left(i\right)}\right)}^{T}\left(Q_{l-1}^{\left(k\right)}+P_{q}^{\left(k\right)}\right)/\sqrt{d_{k}}\right)}$$

This update mechanism allows each pixel to selectively incorporate query information based on relevance, preserving local details such as edges and textures. Particularly for pixels in boundary regions of objects, as they may simultaneously have high similarity with multiple instances, this weighted fusion can more precisely represent boundary transitions, improving segmentation boundary accuracy. The experiment in Section 3.5 shows that, compared to single-pixel updates, the dual-path strategy improves AP by 7.4 percentage points (from 32.5% to 39.9%), demonstrating the significant contribution of this update mechanism to detail capture.

**(2) Query Updates Enhance Global Semantic Understanding:** In the M-step, query features are updated through masked attention and self-attention mechanisms. Masked attention focuses on foreground regions, calculated as follows:(17)$$Q_{l}^{\prime \left(j\right)}=\sum_{i=1}^{HW}\beta_{ji}X_{l}^{\left(i\right)}$$
where $\beta_{ji}$ combines attention similarity and mask information:(18)$$\beta_{ji}=\frac{\exp({\left(Q_{l-1}^{\left(j\right)}+P_{q}^{\left(j\right)}\right)}^{T}\left(X_{l}^{\left(i\right)}+P_{x}^{\left(i\right)}\right)/\sqrt{d_{k}}+M_{l-1}^{\left(j,i\right)})}{\sum_{k=1}^{HW}\exp\left({\left(Q_{l-1}^{\left(j\right)}+P_{q}^{\left(j\right)}\right)}^{T}\left(X_{l}^{\left(k\right)}+P_{x}^{\left(k\right)}\right)/\sqrt{d_{k}}+M_{l-1}^{\left(j,k\right)}\right)}$$

Masked attention allows queries to focus on the foreground regions of their corresponding instances, enhancing the category perception capability. The subsequent self-attention mechanism enables information exchange between queries, building global semantic understanding:(19)$$Q_{l}^{\prime \prime \left(j\right)}=\sum_{k=1}^{N}\gamma_{jk}Q_{l}^{\prime \left(k\right)}$$
where $\gamma_{jk}$ represents the attention weight between query *j* and query *k*:(20)$$\gamma_{jk}=\frac{\exp({\left(Q_{l}^{\prime \left(j\right)}+P_{q}^{\left(j\right)}\right)}^{T}\left(Q_{l}^{\prime \left(k\right)}+P_{q}^{\left(k\right)}\right)/\sqrt{d_{k}})}{\sum_{m=1}^{N}\exp\left({\left(Q_{l}^{\prime \left(j\right)}+P_{q}^{\left(j\right)}\right)}^{T}\left(Q_{l}^{\prime \left(m\right)}+P_{q}^{\left(m\right)}\right)/\sqrt{d_{k}}\right)}$$

#### 2.5.2. Implementation Details

The specific implementation of the dual-path decoder includes four key components:

**(1) Position Embedding:** A learnable position embedding matrix $P\in R^{S\times S\times 256}$ is employed, where S is determined by the IA-guided query number $N_{a}$. The P matrix is interpolated to match the dimensions of feature maps $E3^{\prime }$ and $E4^{\prime }$:(21)$$P_{x}=\mathrm{Interpolate}\left(P,\;\mathrm{size}\left(E3^{\prime }\right)\right),\;P_{q}=\mathrm{Extract}\left(P,\;\mathrm{locations}\;\mathrm{of}\;\mathrm{IA}-\mathrm{guided}\;\mathrm{queries}\right)$$

Additionally, $N_{b}$ learnable position embeddings are used to enhance the discrimination of complex backgrounds.

**(2) Pixel Feature Update:** To improve computational efficiency, cross-attention mechanisms are used instead of self-attention:(22)$$X_{l}=\mathrm{FFN}\left(\mathrm{CrossAttention}\left(X_{l-1}+P_{x},\;Q_{l-1}+P_{q},\;Q_{l-1}+P_{q}\right)\right)$$

The cross-attention layer facilitates query-pixel information fusion, incorporating position embeddings to enhance spatial sensitivity and object distinction.

**(3) Query Update:** An asymmetric design is adopted, implementing masked attention, self-attention, and feed-forward networks:(23)$$\begin{array}{cc} Q_{l}^{\prime } & =\mathrm{MaskedAttention}(Q_{l-1}+P_{q},\;X_{l}+P_{x},\;X_{l}+P_{x},\;M_{l-1})\\ Q_{l}^{\prime \prime } & =\mathrm{SelfAttention}(Q_{l}^{\prime }+P_{q},\;Q_{l}^{\prime }+P_{q},\;Q_{l}^{\prime }+P_{q})\\ Q_{l} & =\mathrm{FFN}(Q_{l}^{\prime \prime }) \end{array}$$

Masked attention focuses queries on foreground regions, self-attention implements comprehensive interaction between queries, position embeddings enhance spatial understanding, and the feed-forward network completes feature integration.

**(4) Prediction Generation:** Each decoder layer contains two independent three-layer MLPs for refining IA-guided queries:(24)$$C_{l}=\mathrm{Softmax}\left({\mathrm{MLP}}_{\mathrm{cls}}\left(Q_{l}\right)\right),\;E_{l}={\mathrm{MLP}}_{\mathrm{mask}}\left(Q_{l}\right)$$

The first MLP predicts object categories (including a “no object” category) while the second generates mask embeddings that encode spatial information. Dynamic mask generation employs layer-specific linear projections:(25)$$M_{l}^{\left(j,i\right)}=E_{l}^{\left(j\right)}\cdot X_{l}^{\left(i\right)}$$

Final confidence scores are derived from category probabilities and segmentation mask scores.

### 2.6. Loss Function

The loss function in this section consists of two parts, which are mainly used for the auxiliary classification header and the two-path Transformer decoder; the former requires training to learn prediction and generation, and the latter requires training to constantly update the object query. The overall loss function of this network is as follows:(26)$$L=L_{IA}+L_{pre}$$
which is the instance activation loss of the auxiliary classification header for the $L_{\mathrm{IA}}$ instance-activation-guided query, and $L_{\mathrm{pre}}$ is the predicted loss instance activation loss. $L_{\mathrm{IA}}$ is defined as follows:(27)$$L_{IA}=\lambda_{cls}\cdot L_{cls}+\lambda_{loc}\cdot L_{loc}$$
where $L_{cls}$ is the category prediction loss, $L_{loc}$ is the location matching cost, and $\lambda_{cls}$ and $\lambda_{loc}$ are hyperparameter balancing factors used to balance the classification loss and mask loss. For $L_{pre}$, we use the Hungarian algorithm to search for the optimal bipartite matching between the prediction set and the ground truth set. This matching strategy maximizes the IoU between the prediction set and the ground truth set, thereby improving segmentation accuracy. Following MaskFormer [11], $L_{pre}$ is defined as follows:(28)$$L_{pre}=\sum_{i=0}^{D}\left(\lambda_{ce}L_{ce}^{i}+\lambda_{dice}L_{dice}^{i}\right)+\lambda_{cls}L_{cls}^{i}$$

In the context of the Transformer decoder, the variable D represents the number of layers. The value of i = 0 denotes the prediction loss of the IA bootstrap query prior to its integration into the Transformer decoder. The variables $L_{ce}^{i}$ and $L_{dice}^{i}$ refer to the binary cross-entropy loss and dice loss, respectively, associated with the segmentation mask. $L_{cls}$ represents the cross-entropy loss for object categorization, with a “no object” weight of 0.1. $\lambda_{ce}$, $\lambda_{dice}$, and $\lambda_{cls}$ are hyperparameters that serve to balance the three losses. Similarly, the Hungarian algorithm is employed to search for the optimal two-part match for target assignment. Additionally, an additional location cost, $\lambda_{loc}$
$L_{loc}$, is incorporated for each query in order to account for the cost of the location of the query.

## 3. Experiments

### 3.1. Specific Realizations

#### 3.1.1. Datasets and Evaluation Indicators

This paper selects the widely recognized Cityscapes and MS COCO Panoptic 2017 datasets in the field of panoptic segmentation as the basis for model training and evaluation. The Cityscapes dataset primarily focuses on urban street scenes and contains 5000 high-quality annotated images (2975 for training, 500 for validation, and 1525 for testing), with an image resolution of 1024 × 2048 pixels. This dataset contains 19 semantic categories, with instance-level annotations provided for 8 of these categories. The MS COCO Panoptic 2017 dataset contains images from common scenes, with a maximum image resolution of 640 × 640 pixels. This dataset contains 133 semantic categories (80 of which are instance categories), and the training, validation, and test sets comprise 118 k, 5 k, and 20 k images, respectively. Both datasets are representative benchmarks in the field of panoptic segmentation and enable effective evaluation of the model’s performance across diverse scenes.

The standard evaluation metric for panoptic segmentation is PQ, which combines segmentation quality (SQ) and recognition quality (RQ). SQ measures pixel-level segmentation accuracy while RQ evaluates object identification and classification accuracy. Specifically, SQ represents the mean IoU of correctly recognized object masks with an IoU exceeding 0.5. The metrics PQ, SQ, and RQ are defined in Equation (Equation 29), where TP denotes correctly predicted instances, FP indicates incorrectly predicted instances, and FN represents cases where categories were correctly identified but instances were incorrectly identified. PQ values range from 0 to 1 and are typically expressed as percentages.(29)$$\begin{array}{cc} PQ & =SQ\times RQ\\ \; & =\frac{\sum_{(p,\;g)\in TP}IoU\left(p,g\right)}{\left|TP\right|+\frac{1}{2}\left|FP\right|+\frac{1}{2}\left|FN\right|} \end{array}$$

PQ not only integrates the performance of the model in both instance segmentation and semantic segmentation but is further refined into two sub-metrics, $PQ^{st}$ and $PQ^{th}$. Among these metrics, $PQ^{st}$ is used to assess the quality of instance-free class segmentation (typically the background) while $PQ^{th}$ is employed to evaluate instance-based class segmentation (foreground objects). This refined evaluation system enables researchers to analyze the model’s performance in diverse scenarios with greater comprehensiveness and precision.

#### 3.1.2. Implementation Details

In the present study, we selected the current widely used and most representative static query Mask2Former as the basis for our investigation. We then proceeded to integrate CMBA, instance bootstrap activation, and dual-path Transformer decoder modules into this framework. With regard to the configuration of the model, Res50 is employed as the underlying network. The parameter batch size was set to 16, the learning rate to 0.0001, and the number of training iterations to 90,000 using the AdamW optimizer [17]. The experiments described in this paper were conducted on a single GPU, utilizing a GPU model NVIDIA Tesla A100 with 40 GB of video memory, the deep learning framework Pytorch 2.1.0, and CUDA version 12.1.

### 3.2. Experimental Results and Analysis

To validate the effectiveness of the proposed method, we compare it with various approaches, including top-down, bottom-up, and static-query-based methods. We selected the commonly used and representative panoptic segmentation metrics described in the previous section: PQ, $PQ^{th}$S, and $PQ^{st}$. The panoptic segmentation results on the Cityscapes val and MS COCO Panoptic 2017 val datasets are presented in Table 1 and Table 2, respectively. The results demonstrate that the proposed panoptic segmentation network, based on dynamic instance queries, outperforms all other methods under the same backbone network configuration.

On the Cityscapes val dataset, PSM-DIQ achieved 63.9%, 56.8%, and 69.3% on the PQ, $PQ^{th}$, and $PQ^{st}$ metrics, respectively. These results represent improvements of 1.8, 2.0, and 2.0 percentage points, respectively, over the current state-of-the-art static query method, Mask2Former.

On the MS COCO Panoptic 2017 val dataset, PSM-DIQ achieved PQ, $PQ^{th}$, and $PQ^{st}$ values of 53.6%, 59.6%, and 44.5%, respectively. Compared to the current state-of-the-art static query method, Mask2Former, these represent improvements of 1.7, 1.9, and 1.5 percentage points, respectively. PSM-DIQ also outperformed all listed top-down, bottom-up, and static-query-based methods.

Figure 5 shows how the total loss decreases during the training process across 80,000 iterations. Key observations: the loss starts very high (around 100) and drops dramatically in the early iterations. There is a steep decline in the first few thousand iterations, indicating rapid initial learning. After about 40,000 iterations, the loss continues to decrease but at a much slower rate. By 80,000 iterations, the loss appears to have almost fully converged to a stable minimum. This pattern is typical of successful deep learning training, showing proper convergence without obvious issues like divergence or significant oscillations.

Figure 6 shows the panoptic quality (PQ) metric on the validation set throughout training: PQ starts at around 40% and shows steady improvement. There is a noticeable jump in performance in the early stages (up to 20,000 iterations). The metric reaches approximately 55–60% by 20,000 iterations. The performance continues to improve but at a slower rate thereafter. Slight fluctuations in the validation PQ are normal and expected. The training process is stable without signs of overfitting (as the validation PQ continues to improve). Overall, the PQ metric indicates that the model is learning effectively and improving its performance on the validation set over time.

### 3.3. Visual Results Analysis

To demonstrate the effectiveness of our method in processing complex visual sensor data, we conducted comprehensive experiments on challenging outdoor scenes. We randomly selected three complex urban environments from the Cityscapes and COCO panoptic segmentation datasets, which contain high-resolution sensor data captured by vehicle-mounted cameras in real-world driving scenarios. The visualization results use different colors to distinguish each segment. From left to right, the images presented are the raw sensor input, the ground truth, the Mask2Former prediction, and the PSM-DIQ (ours) prediction.

Analysis of cityscape dataset results (Figure 7): Small target detection capability: as can be seen from the red circle area, PSM-DIQ can more accurately identify and segment small targets such as pedestrians, traffic signs, and street lights on the road, while Mask2Former has obvious omissions in these areas. Edge accuracy: PSM-DIQ is more accurate in processing object edges, especially at the junction of vehicles and backgrounds, buildings, and the sky. Scene coherence: PSM-DIQ’s segmentation results are more coherent, reducing the fragmented segmentation phenomenon of Mask2Former. Sky region processing: in the first row of images, PSM-DIQ’s segmentation of the sky region is closer to the ground truth, while Mask2Former performs poorly in this area.

Analysis of COCO dataset results (Figure 8): Complex scene understanding: PSM-DIQ also performs well in various off-road scenarios, especially when dealing with diverse environments such as cats, aerial perspectives, and skiing scenes. Detail preservation: the red circled area shows that PSM-DIQ can better preserve the details in the image, such as the skier and small objects in the distance Boundary accuracy: in aerial images, PSM-DIQ recognizes the boundaries of ground objects more accurately, reducing false segmentations.

As shown in Figure 7 and Figure 8, our method demonstrates superior capabilities in processing visual sensor data compared to the baseline network Mask2Former. In these scenes, Mask2Former exhibits limitations in processing critical sensor data elements, including poor segmentation of the sky, street lights, small objects, and object edges, with a considerable loss of detail. These limitations could potentially impact the reliability of downstream sensor-based applications such as autonomous navigation and obstacle detection. In contrast, our method accurately reconstructs the true segmentation, with results closer to the ground truth, making it more suitable for real-world sensor applications.

Overall, PSM-DIQ shows clear advantages in processing visual sensor data, especially in retaining fine-grained sensor information and handling complex urban scenes. These improvements are of great value to fields that rely on high-quality visual perception, such as autonomous driving systems, robotic vision applications, and smart city monitoring. Especially in safety-critical applications, the more accurate scene understanding capabilities provided by PSM-DIQ will greatly improve system reliability.

### 3.4. Ablation Experiments

To verify the contribution of each module in the proposed network, ablation experiments were conducted on the Cityscapes dataset. These experiments provide a deeper analysis of the PSM-DIQ model, focusing on validating the effectiveness of its key components. We designed the following experiments, primarily focusing on three core components: the convolutional block attention module (CBAM), instance activation (IA)-guided queries, and the dual-path update strategy. These components were individually evaluated by systematically removing them from the complete model. For the experimental setup, ResNet-50 was used as the backbone network within the PSM-DIQ model. To comprehensively evaluate model performance, the CBAM attention module, the instance-activation-guided query module, and the dual-path Transformer decoder were incrementally added to the baseline network. The results are shown in Table 3, where “-” indicates that a component was not used in the network structure and “✓” indicates that it was used.

From the experimental data in Table 3, it can be seen that, using the Mask2Former algorithm as the baseline, the resulting PQ was 62.1%. Adding the CBAM in the first control experiment increased the PQ by 0.6 percentage points. This indicates that the image can extract rich spatial and channel features through the CBAM, leading to more effective feature utilization by the network. In the second experiment, adding the instance-activation-guided query module increased the PQ by 0.9 percentage points, with all other metrics also showing improvement. This module allows the queries to be effectively associated with instances in the image, eliminating the need to manually set the number of instance objects and improving the network’s overall accuracy. The third experiment added the instance-activation-guided query module to the first experiment (baseline + CBAM). The results, compared to the first experiment, also showed improvement, with a PQ increase of 0.5 percentage points. This suggests that the network benefits from integrating the rich features extracted by the CBAM into the subsequent stage’s input, thereby improving overall accuracy. The fourth experiment incorporated the dual-path update strategy into the second experiment (baseline + IA). The results, compared to the second experiment, also showed improvement, with a PQ increase of 0.6 percentage points. The network continuously updates and optimizes pixel features through the dual-path updates, thus improving overall accuracy. The fifth experiment combined the CBAM with the dual-path update strategy with the baseline, forming the final network model proposed in this paper. The experimental PQ reached 63.9%. In summary, the results of the aforementioned experiments demonstrate that the proposed method effectively improves the accuracy and performance of panoptic segmentation.

In order to more intuitively demonstrate the contribution of each module to the model performance, we performed a visual analysis of all ablation experiments, as shown in Figure 9. From the visualization results, it can be clearly observed that the complete PSM-DIQ model has a significant advantage in segmentation quality compared to the model lacking certain key components. These visualization results not only verify the conclusions that we have drawn in the quantitative analysis but also intuitively demonstrate the importance of each innovative module from a qualitative perspective, strongly supporting the effectiveness of our method.

By analyzing the results of the ablation experiment visualization (Figure 9), we can make the following observations:

First, judging from the results of the baseline model Mask2Former, although it can roughly capture the main objects in the scene, it is obviously insufficient in processing details. For example, there is blurring at the edges of pedestrians and vehicles, and the recognition of some small objects is incomplete or misclassified.

After adding the CBAM, it can be observed that the segmentation results have improved edge clarity, especially at the junction of the road and the sidewalk, and the depiction of the vehicle outline is more accurate, which proves that the CBAM does enhance the model’s perception of key features through spatial and channel attention mechanisms.

The model with the IA-guided query module demonstrates a better understanding of instance-level objects. As can be seen from the figure, the segmentation quality of foreground objects such as pedestrians and vehicles is significantly improved, and the boundaries are more accurate, which verifies the effectiveness of this module in instance recognition.

The addition of the dual-path update strategy further improves the coherence and consistency of the overall segmentation. It can be noted that, when the model using this strategy processes complex scenes, the transition between categories is more natural and the fragmented segmentation is reduced.

The final complete PSM-DIQ model (combining all modules) visually presents the best segmentation results. It can be clearly seen from the figure that it not only accurately captures the shape and position of the main objects but also retains the details, such as small objects in the distance and complex texture areas. Compared with the ground truth, the results of PSM-DIQ are the closest, which intuitively proves that the various modules that we proposed can produce the best segmentation performance when working together.

These visualization results are highly consistent with the quantitative analysis in Table 3, further confirming the effectiveness of the innovative modules in the PSM-DIQ model, especially in improving the segmentation accuracy of key elements such as pedestrians, vehicles, and road surfaces in urban road scenes.

### 3.5. Extended Experiments

This section presents a series of ablation studies to further analyze the performance and impact of key components in the PSM-DIQ model. The experiments were conducted on the Cityscapes val dataset, using ResNet-50 as the backbone network, the AdamW optimizer, an initial learning rate of 0.0001, a weight decay of 0.05, a batch size of 16, and 90,000 training iterations. Unless otherwise specified, the parameters and environmental configurations are identical to those in Section 3.1.2 The evaluation metrics include panoptic quality (PQ), mean intersection over union (mIoU), floating-point operations (FLOPs), and frames per second (FPS).

The Table 4 results show that, as the number of decoder layers increases from 1 to 3, the AP of both Mask2Former and the IA-guided method significantly increases. This indicates that increasing the number of decoder layers enhances the model’s representational capacity, thereby improving segmentation performance. For the same number of decoder layers, the IA-guided method consistently achieves higher AP than Mask2Former. For example, when D = 1, IA-guided’s AP is 35.6%, 1.0% higher than Mask2Former; when D = 3, IA-guided’s AP is 39.9%, 2.6% higher than Mask2Former. This indicates that the IA-guided strategy can utilize query information more effectively and improve accuracy. Although the IA-guided method improves AP, its FLOPs and FPS are comparable to those of Mask2Former. This indicates that the IA-guided method, while improving segmentation accuracy, does not significantly increase the computational burden, thus maintaining good computational efficiency. Furthermore, the greater the number of decoder layers, the greater the improvement observed with IA-guided. This may be because a deeper decoder can better utilize the high-quality queries provided by IA-guided.

The Table 5 results show that the average precision (AP) of the dual-path update strategy is significantly higher than that of the single-path update strategy. This indicates that simultaneously updating pixel embeddings and query embeddings can utilize information more fully, thereby significantly improving accuracy. The AP achieved with the dual-path update at D = 3 is better than that of the single-query update at D = 6. This indicates that the dual-path update strategy can achieve higher accuracy while reducing the number of decoder layers, thereby improving training efficiency. Although single-pixel and dual-path updates have the same FPS, the FLOPs of the former are much higher than those of the latter. In the dual-path update strategy, the “query → pixel” update order is slightly better than the “pixel → query” update order. This may be because query updates can better utilize the information from the previous layer’s predicted mask.

The Table 6 results show that MSDeformAttn has the highest AP (40.0%), indicating that it has the strongest feature extraction and fusion capabilities. However, its computational cost is significantly higher than that of the other decoders, indicating its higher computational complexity. The FLOPs of the FPN and PPM-FPN are comparable, with the Transformer encoder being slightly higher. The inference speed (FPS) of the FPN and PPM-FPN is comparable, and significantly higher than that of the Transformer encoder and MSDeformAttn. The PPM-FPN achieves a good balance between accuracy, computational complexity, and inference speed, making it a suitable choice.

## 4. Conclusions

This paper addresses the limitations of static query mechanisms in visual sensor data processing and proposes a panoptic segmentation network based on dynamic instance queries (PSM-DIQ). Our method enhances sensor data processing performance through three key innovations:

(1) Introduction of the CBAM to fuse spatial and channel information, significantly improving the feature extraction capabilities for visual sensor data;

(2) Selection of high-level semantic pixel embeddings from the sensor’s underlying feature map as queries, enhancing query efficiency and its correlation with sensor-captured content;

(3) Implementation of a dual-path approach that alternates between updates of sensor-derived pixel features and queries, optimizing pixel–query interaction for the accurate capture of image details and semantic information.

Experimental results on the Cityscapes and MS COCO datasets, which contain diverse real-world sensor data, demonstrate that, with a ResNet-50 backbone, PSM-DIQ improves the PQ value by 1.8 and 1.7 percentage points, respectively, compared to Mask2Former, validating its effectiveness in processing visual sensor information.

### Applications in Autonomous Driving and Robotics

The PSM-DIQ model offers significant potential for autonomous driving applications through its enhanced segmentation capabilities:

(1) The dynamic query mechanism improves critical object recognition in complex traffic scenarios, particularly for distant pedestrians, cyclists, and traffic signs that are essential for collision avoidance systems.

(2) The dual-path update strategy enables more precise drivable area segmentation, supporting lane-keeping and path-planning algorithms with more accurate environmental boundaries.

(3) The enhanced processing of small objects and boundary details contributes to more reliable obstacle detection, which is crucial for emergency braking and trajectory adjustment systems.

In the field of robot vision, our approach offers several advantages:

(1) Precise instance segmentation enables robots to distinguish between multiple similar objects in cluttered environments, facilitating accurate grasping and manipulation tasks.

(2) Improved semantic understanding supports spatial reasoning capabilities for indoor navigation and human–robot interaction scenarios.

(3) The flexible query mechanism adapts to varying object counts in dynamic environments, making it suitable for robot perception in changing scenes.

## Figures and Tables

**Figure 1 sensors-25-02919-f001:**
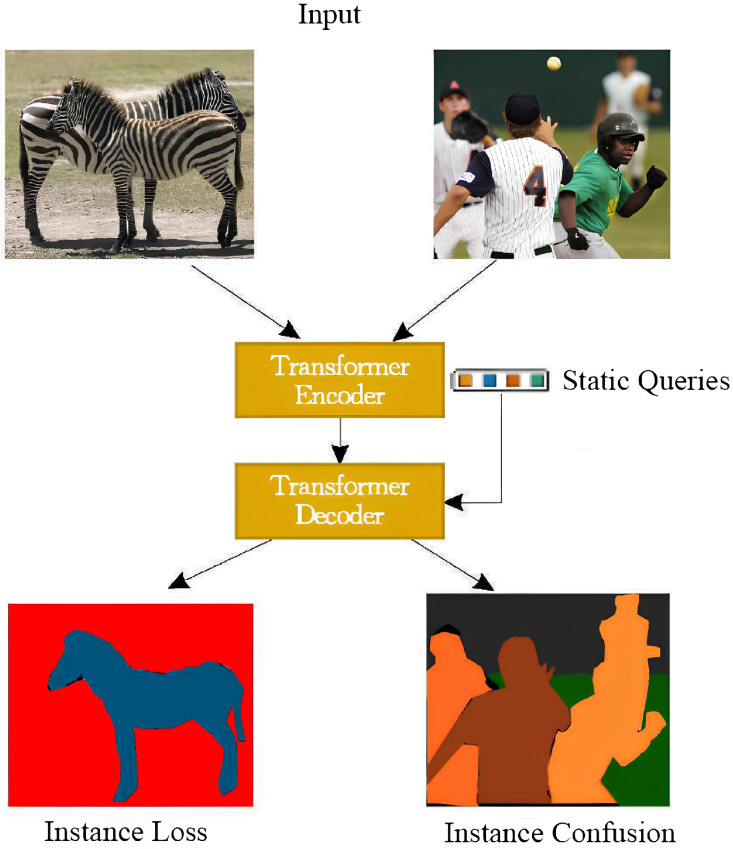
Problems with “static” query segmentation results.

**Figure 2 sensors-25-02919-f002:**
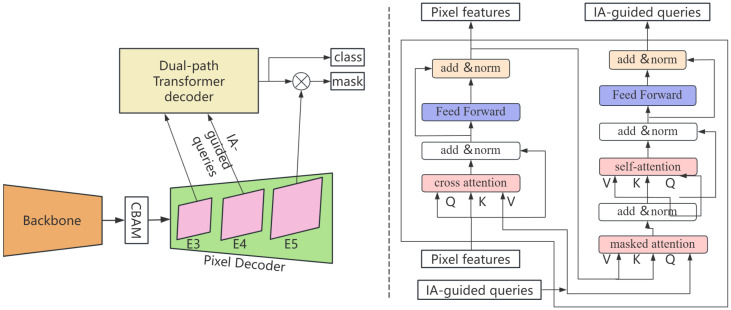
Frame diagram of PSM-DIQ panoptic method.

**Figure 3 sensors-25-02919-f003:**
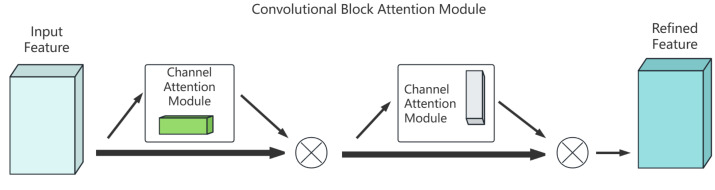
Frame diagram of PSM-DIQ panoptic method.

**Figure 4 sensors-25-02919-f004:**
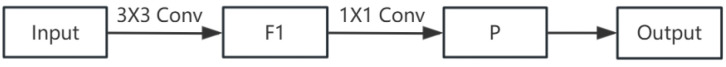
Auxiliary classification head structure.

**Figure 5 sensors-25-02919-f005:**
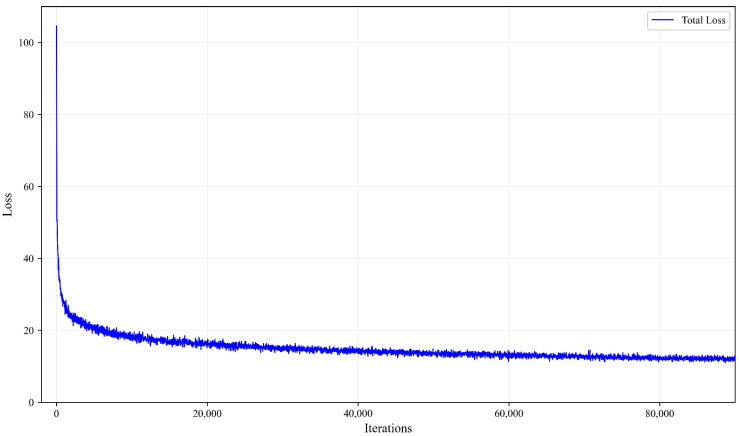
Loss curve of the proposed method during training.

**Figure 6 sensors-25-02919-f006:**
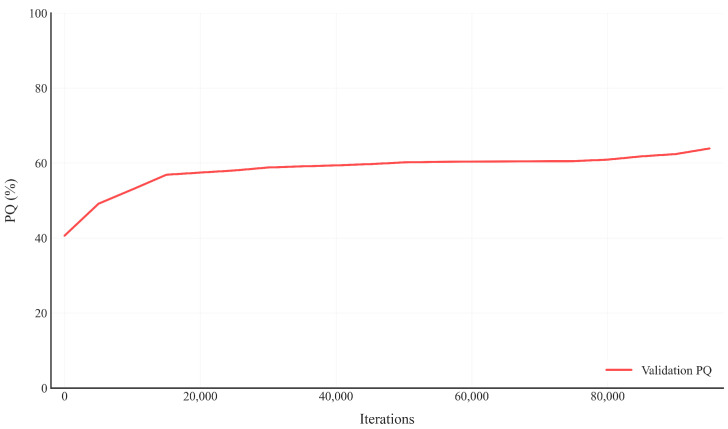
PQ curve of the proposed method on the Cityscapes val dataset.

**Figure 7 sensors-25-02919-f007:**
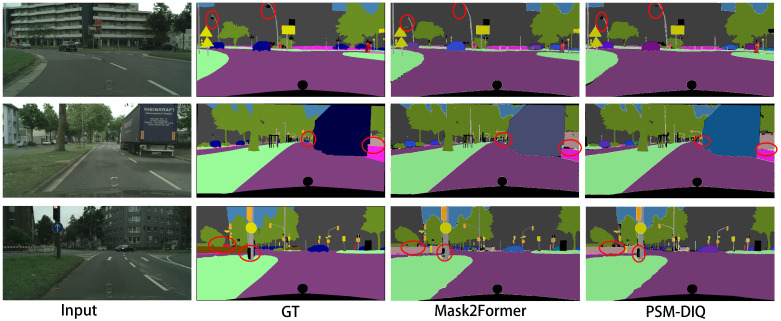
Example of a cityscape validation set prediction.

**Figure 8 sensors-25-02919-f008:**
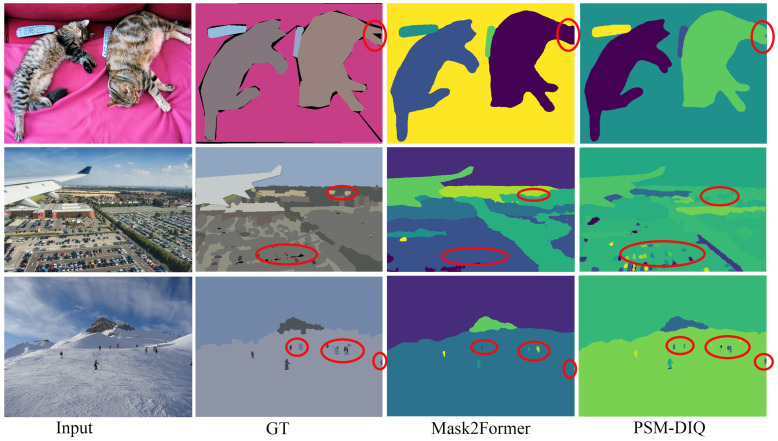
Example of a COCO validation set prediction.

**Figure 9 sensors-25-02919-f009:**
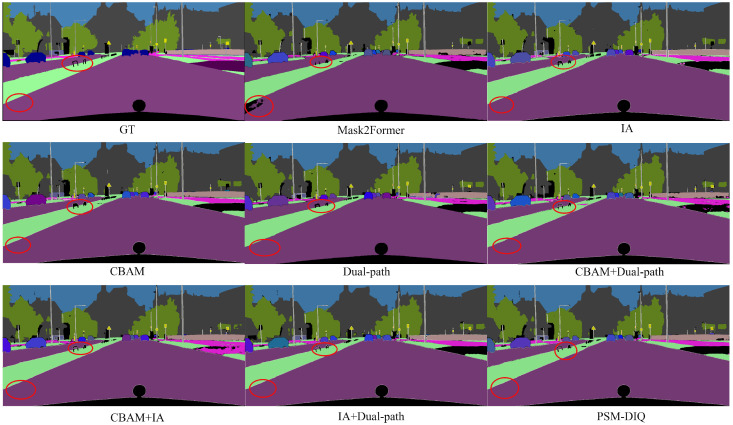
Ablation study results.

**Table 1 sensors-25-02919-t001:** Comparison of panoptic segmentation results on the Cityscapes val dataset.

Method	Backbone	PQ	PQ^th^	PQ^st^
*Top-down Approach*				
Panoptic FPN [5]	Res101	58.1	52.0	62.5
RT Panoptics [18]	Res50	58.8	52.1	63.7
AUNet [19]	Res101	59.0	54.8	62.1
UPSNet [6]	Res50	59.3	54.6	62.7
SE-PSNet [20]	Res50	60.0	55.9	62.9
EfficientPS [21]	Res50	60.3	55.3	53.9
CCPSNet [22]	Res50	60.5	56.9	63.1
Unilying [23]	Res50	61.4	54.7	66.3
*Bottom-up Approach*				
Deeperlab [8]	Xception-71	56.5	—	—
SSAP [24]	Res50	58.4	50.6	—
AdaptIS [25]	Res50	59.0	55.8	61.3
Panoptic-DeepLab [7]	Res50	60.3	51.1	67.0
*Static Query Methods*				
PanopticFCN [26]	Res50	61.4	54.8	66.6
Mask2Former [11]	Res50	62.1	54.8	67.3
*Dynamic Query Methods*				
K-Query [27]	Res50	63.2	56.2	68.3
**PSM-DIQ**	**Res50**	**63.9**	**56.8**	**69.3**

**Table 2 sensors-25-02919-t002:** Comparison of panoptic segmentation results on the MS COCO val dataset.

Method	Backbone	PQ	PQ^th^	PQ^st^
*Top-down Approach*				
Panoptic FPN [5]	Res101	39.0	45.9	28.7
AUNet [19]	Res101	39.6	49.1	25.2
UPSNet [6]	Res50	42.5	48.6	33.4
Unilying [23]	Res50	43.4	48.6	35.5
*Bottom-up Approach*				
SSAP [24]	Res50	36.5	—	—
Panoptic-DeepLab [7]	Res50	35.5	37.8	32.0
IDNet [28]	Res50	42.1	47.5	33.9
CCPSNet [22]	Res50	43.0	49.2	33.6
*Static Query Methods*				
PanopticFCN [26]	Res50	44.3	50.0	35.6
K-Net [29]	Res50	47.1	51.7	40.3
Max-DeepLab [30]	Max-X	48.4	53.0	41.5
Mask2Former [7]	Res50	51.9	57.7	43.0
*Dynamic Query Methods*				
K-Query [27]	Res50	52.9	58.9	43.8
**PSM-DIQ**	**Res50**	**53.6**	**59.6**	**44.5**

**Table 3 sensors-25-02919-t003:** Results of set ablation experiments on the dataset Cityscapes val.

Method	Backbone	CBAM	IA-Guided	Dual-Path	PQ	PQ^th^	PQ^st^	AP	IoU
Mask2Former	Res50	-	-	-	62.1	54.8	67.3	37.3	77.5
		✓	-	-	62.7	54.8	68.6	36.8	78.5
PSM-DIQ		-	✓	-	63.0	55.6	68.4	38.0	78.9
		-	-	✓	62.7	54.8	67.9	37.7	77.8
		✓	✓	-	63.2	55.8	68.6	38.6	79.6
		✓	-	✓	63.3	55.4	69.0	38.0	79.0
		-	✓	✓	63.6	56.0	69.2	38.4	78.8
		✓	✓	✓	**63.9**	**56.8**	**69.3**	**39.9**	**80.0**

**Table 4 sensors-25-02919-t004:** Performance of instance-activation-guided query under different decoders.

Network	D	AP	FLOPs	FPS
Mask2Former	1	34.6	58.4 G	50.0
	3	37.3	74.3 G	36.1
IA-guided	1	**35.6**	59.6 G	48.8
	3	**39.9**	75.5 G	35.5

**Table 5 sensors-25-02919-t005:** Performance of dual-path update strategy under different decoders.

Method	D	AP	FLOPs	FPS
Single-pixel update	6	32.5	85.4 G	35.5
Single-query update	6	36.9	63.3 G	35.0
Dual-path (pixel–query)	3	**37.8**	75.5 G	35.5
Dual-path (query–pixel)	3	**39.9**	75.5 G	35.5

**Table 6 sensors-25-02919-t006:** Selection of pixel decoder.

Method	AP	FLOPs	FPS
FPN	37.4	75.4 G	35.7
Transformer Encoder	38.9	78.5 G	29.5
MSDeformAttn	40.0	114.7 G	21.2
PPM-FPN	39.9	75.5 G	35.5

## Data Availability

Data are contained within the article.

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
