# Peer review of "Panoptic Image Segmentation Method Based on Dynamic Instance Query"

_sensors, 2025, doi:10.3390/s25092919_

Round 1
Reviewer 1 Report
Comments and Suggestions for Authors
The paper is well-written with a sufficient number of examples. It has practical and scientific value. However, there are some points that require clarification.
- I recommend expanding the introduction section with literature survey as related works haven’t been included in the manuscript.
- Enhance your proposed work with mathematical equation in subsections 2.2-2.5 where possible
- Try to add graphical results of the training and testing, (Loss, etc)
- Visualize architecture of auxiliary classification head structure in subsection 2.4
- Can you explain how spatial and channel information are fused? Provide mathematical equation in the paper also.
- Regarding the dual-path Transformer decoder, How can switching between updating query features and pixel features help to improve image detail and semantic data capture? Discuss in the paper also.
Reviewer 2 Report
Comments and Suggestions for Authors
This paper proposes a novel panoptic segmentation method based on dynamic instances called PSM-DIQ. It is a valuable study, and much work has been done, but there are still some issues and content that need to be revised or explained.
- The introduction provides a comprehensive review of existing work but lacks depth in discussing the limitations of current methods.
- The methodology section does not clearly describe the specific implementation details of the Dynamic Instance Query. It is suggested to include more technical details, especially on how dynamic queries are generated from the underlying feature.
- The description of the dual-path Transformer decoder is somewhat abstract. Providing more mathematical formulations or pseudocode would help readers better understand its working principles.
- The experimental section lacks comparisons with other dynamic query methods to further validate the superiority of PSM-DIQ.
- More ablation studies could be added, particularly to analyze the independent contributions of the dynamic query generation mechanism and the dual-path decoder.
- The analysis of the reasons behind these improvements in results is not thorough. It is suggested to further analyze the advantages of PSM-DIQ in handling complex scenes, especially in terms of small object and edge detail processing.
- Additional visual results could be included to demonstrate the specific performance of PSM-DIQ in complex scenes, particularly in addressing instance loss and confusion issues.
- The discussion section could be expanded to explore the potential applications in autonomous driving and robot vision.
The English could be improved to more clearly express the research.
Round 2
Reviewer 1 Report
Comments and Suggestions for Authors
The authors have addressed all my comments. I recommend accepting the paper in its current format.
Author Response
Comment 1:The authors have addressed all my comments. I recommend accepting the paper in its current format.
Response 1:Thank you for your positive comments on our paper and your valuable comments and suggestions during the review process. These comments have played a positive role in improving the quality of the paper.
Thank you again for recommending your paper to be accepted. We are deeply honored and will continue to maintain a rigorous attitude in subsequent research.
Reviewer 2 Report
Comments and Suggestions for Authors
The authors have made significant revisions based on feedback, but there are still some issues that need to be improved and addressed.
(1) In the authors' responses, there may be numerous traces of AI tools, such as ChartGPT or DeepSeek.
(2) More visualization results in comparative and ablation experiments will increase credibility。
(3) The logical coherence of the manuscript needs to be strengthened, rather than simple statements as in section 2.1. Network Infrastructure.
Comments on the Quality of English LanguageThe English could be improved to more clearly express the research.
